# Development and validation of risk prediction model for diabetic neuropathy among diabetes mellitus patients at selected referral hospitals, in Amhara regional state Northwest Ethiopia, 2005–2021

**Negalgn Byadgie Gelaw** [1]*, **Achenef Asmamaw Muche**[2], **Adugnaw Zeleke Alem**[2], **Nebiyu Bekele Gebi**[3], **Yazachew Moges Chekol**[4], **Tigabu Kidie Tesfie** [5], **Tsion Mulat Tebeje** [6]

1 Department of Public Health, Mizan Aman College of Health Sciences, Mizan-Aman, Ethiopia,
2 Department of Epidemiology and Biostatistics, Institute of Public Health, College of Medicine and Health Sciences, University of Gondar, Gondar, Ethiopia, 3 Department of Internal Medicine, School of Medicine, University of Gondar Comprehensive Specialized Hospital, Gondar, Ethiopia, 4 Department of Health Information Technology, Mizan Aman College of Health Sciences, Mizan-Aman, Ethiopia, 5 Department of Public Health, College of Health Sciences, Debre Markos University, Debre Markos, Ethiopia, 6 Unit of Epidemiology and Biostatistics, School of Public Health, College of Medicine and Health Sciences, Dilla University, Dilla, Ethiopia

* negalgnbyadgie21@gmail.com

## Abstract

### Background

Diabetic neuropathy is the most common complication in both Type-1 and Type-2 DM patients with more than one half of all patients developing nerve dysfunction in their lifetime. Although, risk prediction model was developed for diabetic neuropathy in developed countries, It is not applicable in clinical practice, due to poor data, methodological problems, inappropriately analyzed and reported. To date, no risk prediction model developed for diabetic neuropathy among DM in Ethiopia, Therefore, this study aimed prediction the risk of diabetic neuropathy among DM patients, used for guiding in clinical decision making for clinicians.

### Objective

Development and validation of risk prediction model for diabetic neuropathy among diabetes mellitus patients at selected referral hospitals, in Amhara regional state Northwest Ethiopia, 2005–2021.

### Methods

A retrospective follow up study was conducted with a total of 808 DM patients were enrolled from January 1,2005 to December 30,2021 at two selected referral hospitals in Amhara regional state. Multi-stage sampling techniques were used and the data was collected by checklist from medical records by Kobo collect and exported to STATA version-17 for analysis. Lasso method were used to select predictors and entered to multivariable logistic

**Data Availability Statement:** All relevant data are within the paper and its Supporting information files.

**Funding:** We greatly acknowledge Mizan Aman College of health sciences for sponsorship.

**Competing interests:** The authors have declared that no competing interests exist.

**Abbreviations:** AUC, Area under the Curve; BMI, Body Mass Index; CI, Confidence Interval; CRT, Classification and Regression Tree; DBP, Diastolic Blood Pressure; DCA, Decision Curve Analysis; DM, Diabetes Mellitus; DPN, Diabetic Peripheral Neuropathy; FBG, Fasting Blood Glucose; HDL, High Density Lip-Protein; HbA1c, Glycated Hemoglobin A1 c; ICT, Information Communication Technology; IDF, International Diabetes Federation; IQR, Inter Quartile Range; LASSO, Least Absolute Shrinkage and Selection Operator; LDL, Low Density Lip-Protein; LR+, Log Likelihood Ratio Positive; LR_, Log Likelihood Ratio Negative; MNSI, Michigan Neuropathy Screening Instrument; NPV, Negative Predictive Value; PBG, Post Bradial Glucose; PPV, Positive Predictive Value; RBC, Red Blood Cells; ROC, Receiver Operating Characteristics; SBP, Systolic Blood Pressure; SRMA, Systematic Review and Meta-Analysis; T1DM, Type-1 Diabetes Mellitus; T2DM, Type-2 Diabetes Mellitus; TC, Total Cholesterol; TG, Total Triglyceride; UOG, University of Gondar; WBC, White Blood Cells.

regression with P-value<0.05 was used for nomogram development. Model performance was assessed by AUC and calibration plot. Internal validation was done through bootstrapping method and decision curve analysis was performed to evaluate net benefit of model.

## Results

The incidence proportion of diabetic neuropathy among DM patients was 21.29% (95% CI; 18.59, 24.25). In multivariable logistic regression glycemic control, other comorbidities, physical activity, hypertension, alcohol drinking, type of treatment, white blood cells and red blood cells count were statistically significant. Nomogram was developed, has discriminating power AUC; 73.2% (95% CI; 69.0%, 77.3%) and calibration test (P-value = 0.45). It was internally validated by bootstrapping method with discrimination performance 71.7 (95% CI; 67.2%, 75.9%). It had less optimism coefficient (0.015). To make nomogram accessible, mobile based tool were developed. In machine learning, classification and regression tree has discriminating performance of 70.2% (95% CI; 65.8%, 74.6%). The model had high net benefit at different threshold probabilities in both nomogram and classification and regression tree.

## Conclusion

The developed nomogram and decision tree, has good level of accuracy and well calibration, easily individualized prediction of diabetic neuropathy. Both models had added net benefit in clinical practice and to be clinically applicable mobile based tool were developed.

## Background

Diabetes mellitus is metabolic disorder disease characterized by increasing blood glucose level in the body resulting from either defect insulin secretion, insulin action, or both [1]. Globally, around 3 million deaths annually is due to DM and 49.7% living with diabetes undiagnosed [2]. It is one of the most common problem in the world has faced both in developed and developing countries [3]. In Africa, by 2019, around 19 million adult populations were estimated to have diabetes and it is expected that by 2045 it will be around 47 million and similarly in Ethiopia, an estimated 1,699,400 adults were living with diabetes mellitus [4].

The chronic hyperglycemia of diabetes mellitus had long-term damage, dysfunction, and failure of different organs such as nerves, eyes, kidneys [5]. Diabetic neuropathy defined as presence of symptoms and signs of peripheral nerve dysfunction in people with diabetes mellitus after exclusion of other causes [6]. The mechanism of nerve dysfunctions was through metabolic process by hyperglycemia due to oxidative stress, directly affecting nerve fibers [7]. It is the most common complication of both type-1 and type-2 DM with more than one half of all patients developing nerve dysfunction in their lifetime [8].

The Burdon of diabetic neuropathy common in industrialized countries ranges from in china, 33.1% [9] to 40.3% among patients with type-1 diabetes mellitus and 42.2% with type-2 diabetes mellitus in Germany [10]. Evidence from previous study documented shows that in Latin American and Caribbean countries, the incidence of diabetic neuropathy was 29.2% [11] and in Sri Lankan 28.8% [12].

In Africa, according to a systematic review and meta-analysis done, the overall pooled prevalence of diabetic neuropathy was 46% [13]. In south Africa 30.3% [14], in Ghana 8.1% [15], in Northern Africa, incidence ranges from 21.9% to 60% [16].

In Ethiopia, the prevalence of diabetic neuropathy ranges from 16.63% at university of Gondar comprehensive specialized referral hospital to 52.2% at Bahir-Dar city [17], at Tikur-Anbesa and St. Paul's specialized university hospital was 48.2% with 53.6% in type 2 and 33.3% in type 1 diabetes mellitus patients [18].

As part of NCD prevention and treatment strategies, diabetic neuropathy and related complications are getting attention at the national and international levels [19]. Guidelines are recommending that by multidisciplinary team approach can treating and preventing diabetic neuropathy. WHO and other professional society guideline are also recommended that strengthening of the surveillance & management of diabetic neuropathy was important to reduce the incidence and its consequence [20].

Despite different strategies and initiatives implemented to reduce the Burdon of diabetic neuropathy, up to one third of patients with diabetes have neuropathic pain [21] which often leads to sleep disturbance, poor quality of life, depression, and unemployment. Both acute and chronic diabetic neuropathies had occur in DM patients [22] that results diabetic foot disease, including ulceration and non-traumatic amputations that might be due to lack of screening tool that can be used to identify diabetic patients who is at risk of diabetic neuropathy.

Although, risk prediction model for diabetic neuropathy among DM patients in developed countries was developed, it is not applicable in clinical practice due to poor data, methodological problems, inappropriately analyzed and poorly reported [23]. Delay and lack of detection of the complication was mostly resulted from patients being asymptomatic during the early stage of the disease [24]. So that a simple and accurate risk prediction tool to identify those at high risk of patents develop to diabetic neuropathy had great value.

Early identification of high risk DM patients using simple screening tool, avoids late diagnosis of diabetic neuropathy that reduces huge economic costs and serious complications [25]. Thus, a method used to provide information about their level of risk was needed to take timely intervention measures to prevent occurrence of diabetic neuropathy.

Many epidemiological studies on diabetic neuropathy, including cross-sectional studies are carried out around the world to explore the risk factors associated with diabetic neuropathy [10, 26] but level of risk to develop diabetic neuropathy for DM patients was unknown. Hence studies suggest that patient specific model like individual patients risk prediction model is useful to alleviate the problem.

Risk prediction models have been used to predict the probability of risk of diseases used in medicine and public health to guide clinical decision-making [27]. However, risk prediction model for diabetic neuropathy among DM patients have not been developed in Ethiopia. There was variability in the clinical diagnosis of diabetic neuropathy in different setting [28]. Thus, this prediction model considers prognostic factors that have been generally monitored in clinical practice and precisely measured to ensure its feasibility and accuracy for clinical application. Currently an urgent change for improvement of the diagnosis and management of diabetic neuropathy were needed [29].

According to a previous risk prediction model conducted to identify DM patients at risk of developing diabetic neuropathy being old age [30], sex [31], educational level [32] were significant predictors with area under ROC curve of 0.957. In another study, age [33], residence [34], BMI [35], educational level [32], poor diabetic control [36], type of DM [37], duration of diabetes mellitus [38], number of hypoglycemic drugs used [39] with AUC 0.859 were also determinates of diabetic neuropathy. Hemoglobin level [40], presence of other micro and macro vascular complications [11], hypertension [38], type of DM [31] baseline comorbidities [41], complications such as diabetic nephropathy, diabetic-retinopathy [10], lipid profile (triglyceride, cholesterol) levels [42], fasting blood glucose [43, 44], HbA1c [45], high level of alcohol

drinking [11] physical inactive [38], eating fat containing foods [46] were the most common predictors of diabetic neuropathy.

Although, different strategies and interventions have been made so far, to reduce the Burdon diabetic neuropathy, Overtime at least 50% of individuals with diabetes develop diabetic neuropathy [47]. It is preventable through detected early the high risk patients, by providing timely intervention measures. However, identifying high risk patients for diabetic neuropathy was not performed as expected [24].

Until an advanced stage, diabetic neuropathy is asymptomatic but it can be prevented through identifying high risk patients and prompt treatment before their complication. Up to my knowledge there is no risk prediction model for diabetic neuropathy among DM patients in Ethiopia. As a result, this risk prediction models alleviate the problem by developing nomogram and decision tree, estimate disease risk, can guide healthcare providers for disease intervention and arrangement future health care needs by providing treatment.

Monitoring biomarkers and other predictors is also very essential to be aware of metabolic abnormality by providing meaningful prognostic information that can help to differentiate patients with regard complication and lead the way to change other intervention. This risk prediction model was developed by applying sound statistical methods and analysis to identify DM patients at which level of risk for the development of diabetic neuropathy [48].

Besides, the finding of this study was highly useful as a simple clinical tool to guide clinicians for decision making, aid for specific screening of high-risk patients and for their informed choice of treatment to the patients. Therefore, these studies aimed development and validation of risk prediction model for diabetic neuropathy among diabetes mellitus patients in selected referral hospitals at Amhara regional state, Ethiopia.

## Methods and materials

### Study design and area

An institution-based retrospective follow up study was conducted among patients diagnosed with diabetes mellitus from January 1, 2005 to December 30, 2021. The study area was selected referral hospitals in northwest part of Amara regional state, which includes University of Gondar Comprehensive specialized and referral hospital, Debre markos comprehensive specialized hospital, Tibebe-Gihon comprehensive specialized referral hospital, Felge-Hiwot comprehensive specialized referral hospital, and Debre tabor comprehensive specialized referral hospital.

Using lottery Method, University of Gondar comprehensive specialized referral hospital and Felege-Hiwot comprehensive specialized referral hospital was selected. University of Gondar comprehensive specialized hospital found in Gondar town, far from Addis Abeba, the capital city of Ethiopia 750 km and 200 km from Bahir-dar, the capital city of Amhara regional state. It serves for more than 7 million people in northwest Ethiopia. The hospital serves around 24,862 numbers of people are having chronic follow-up per year, and among this, 4760 were DM patients.

Felege-Hiwot Referral Hospital is found in Bahir-Dar, which is the capital city of Amhara regional state located at 565 km from Addis Ababa, Northwest Ethiopia. It serves as over 7 million people from the surrounding area. Around 21,218 people had a chronic follow-up in this hospital and among these 4200 were DM patients. The study setting was displayed below.

### Population

The source population was all people diagnosed with diabetes mellitus having a follow up at selected referral hospitals in Amhara regional state, North West Ethiopia and all people having confirmed diabetes mellitus started treatment and follow up at selected referral hospitals in

Amhara regional state with in follow up period were study population. A diabetes mellitus patient who was enrolled from January 1, 2005 to December 30, 2021 with minimum of two years follow up, at university of Gondar comprehensive specialized hospital and Felege-Hiwot comprehensive specialized referral hospital were included. Diabetes mellitus patients whose date of initiation was not recorded and incomplete charts, those DM patients develop diabetic neuropathy at beginning, transferred in and gestational diabetes mellitus was excluded.

## Sample size calculation and sampling technique

In risk prediction model development sample size was calculated through different approaches/methods either considering Minimum Mean Absolute Prediction Error (MAPE) to be minimum and use shrinkage factor to be minimize the issue of over fitted model.

Therefore, by applying the expected average error of predicting the outcome when the developed prediction model is going to be applied to new individuals should be considered, that is called Mean Absolute Prediction Error (MAPE) should be minimum. Sample size also can be calculated using the following formula by taking the prevalence according to a systematic review and meta-analysis study done in Ethiopia, the prevalence of diabetic neuropathy among diabetes mellitus patients was 22% [49].

$$n = \exp\left(\frac{-0.508 + 0.259\ln(\phi) + 0.504\ln(p) - \ln(\text{MAPE})}{0.544}\right) \quad (1)$$

Here $\phi$ -is the proportion of diabetic neuropathy P- is the number of predictors used to predict the diabetic neuropathy and MAPE is the mean absolute prediction error of maximum 5% considered in our case.

$$n = \exp\left(\frac{-0.508 + 0.259\ln(0.22) + 0.504\ln(22) - \ln(0.05)}{0.544}\right) = 475$$

The second method is taking the issue of over fitted model in to consideration in prediction model. We have targeted less than 10% over fitted model for predicting diabetic neuropathy in DM patients. The amount of sample size required to minimize the problem of over fitting should be identified. We consider the following sample size formula.

$$n = \frac{P}{(S-1)\ln\left(1 - \frac{R_{Cs}^2}{S}\right)} \quad (2)$$

$$n = \frac{P}{(S-1)\ln\left(1 - \frac{R_{Cs}^2}{S}\right)} = \frac{22}{(0.9-1)\ln\left(1 - \frac{0.325}{0.9}\right)} = 492$$

From the above different types of approaches for sample size calculation in risk prediction model the largest sample size is 492 and since multi stage sampling technique was used, design effect taken as equal to 1.5, Then 492*1.5 = 738.

Finally by adding 15% contingency because of while during data extraction checklist pre-testing, to check completeness and clarity of checklist, most of the patient medical record number was absent in the medical record room by taking in to consideration such missing charts, adding 10–15% contingency was recommended so, the total minimum sample size was 849.

Multi stage sampling techniques were used by first selecting the referral hospitals in North West part of Amhara regional state. Among these referral hospitals by using lottery method UOG comprehensive specialized referral hospital and Felege-Hiwot comprehensive

specialized referral hospital were selected. Then, study participates were selected by first proportion allocation to each selected referral hospitals through the following proportional allocation formula.

$$ni = \frac{n}{N} * Ni$$

Where n = Total sample size N-total population ni- sample size for each hospital.

Finally the study participates was selected by using computer random generated number through preparing sampling frame by arranging his/her medical record number order.

## Variables

**Dependent variable.** Diabetic neuropathy.

**Independent variable. Socio demographic variables**: Baseline age (year), sex, residence, body mass index.

**Laboratory related variables**: Mean fasting blood glucose level, Mean arterial blood pressure, baseline total cholesterol, baseline triglyceride level, baseline value of WBC, RBC, platelets count, baseline hemoglobin and creatinine level.

**Diagnosis and treatment related variables**; Type of DM, duration of DM, Type of treatment for DM, hypertension, baseline other comorbidities and adherence.

**Behavioral related variables;** Alcohol drinking, unhealthy diet, physical activity.

**Operational definition. Diabetic neuropathy;** It can be either small fiber neuropathy or large fiber neuropathy. Small fiber neuropathy manifested by pain, tingling, paraesthesia and confirmed by pinprick and temperature examination. Large fiber neuropathy is manifested by numb feet and gait ataxia and confirmed by touch sensation by 10g monofilament, vibration sense by biothesiometer and ankle reflex. A patient is considered as having diabetic neuropathy if he/she is diagnosed as diabetic neuropathy on his/her medical record [50].

**Other comorbidities**; if other baseline disease (except hypertension) in DM patients were present such as ischemic heart diseases, stroke, e.t.c.

**Glycemic control**: Patients was classified as per the WHO criteria into:

- Good glycemic control = fasting blood glucose of 80–130 mg/dl

- Poor glycemic control = fasting blood glucose of > 130 mg/dl—— [51]

## Data collection tools and quality control

This risk prediction model was used secondary data source, collected from June 8 to July 10, 2022 using the developed data extraction check lists. The reviewed records were identified by their medical registration number. Patient intake form, follow-up card, and DM registration book were used as data sources. Socio-demographic characteristics, behavioral characteristics, baseline, and follow-up clinical and laboratory data were collected from patient cards. The date that patients start regular follow-up treatment until the end of the study to the confirmation of a final event in the study period was collected. Four trained BSc nursing health professional and two public health officer supervisors were recruited for data collection. Training on the objective of the study and how to retrieve records as per data extraction checklists were given to data collectors and supervisors one day before data collection. As well as, random sample from data extracted was crosschecked for its consistency. The information formats were crosschecked with the source card on the spot, and regular supervisions were done.

## Data processing and analysis

Data was coded and entered to Kobo collect Version v2022.1.2 for clean-up and then exported to Stata version-17 for analysis. Descriptive statistics (frequency, percentage, proportion, and mean with standard deviation, median with IQR) were used to describe the clinical course of diabetic neuropathy diseases. The candidate prognostic factors were selected by using the least absolute shrinkage and selection operator (LASSO) method, and selected variables were entered in to multivariable logistic regression and model reduction was done using log likelihood ratio test >0.15 and finally p-value < 0.05 in model reduction were taken as statistically significant predictors and beta coefficients with 95% CI was reported. Mullti-collinerity was checked by variance inflation factor where VIF > 10 indicate the presence of multi-collinearity.

## Missing data management

Missing variable with value <50%, the missing data was handled by multiple imputation through checking the type of missing data. In this risk prediction model the missing variable was checked by fitting statistical test (logistic regression) found a result, the missing variable was associated with the observed variables, indicates that Missing at random (MAR). The percentage of missing value was done by "mdesc" package by STATA version-17 and if the missing value > = 50%, were not included in our model. From 22 predictors, six of them had complete data: baseline age, sex, residence, type of DM, type of treatment for DM and duration. The remaining missing variables were managed by multiple imputation technique. The missing variables and its percentage value was found in (S1 File).

## Risk prediction model development

In risk prediction model development both the prognostic markers and determinates that predict the diabetic neuropathy was selected from socio-demographic factors, diagnosis and treatment related factors, laboratory related factors and behavioral related factors. The determinants factors included in the multivariable logistic regression analysis were selected based on the results of lasso regression method. The theoretical design was the incidence of diabetic neuropathy at a future time "t" is a function of prognostic determinants ascertained at one time (baseline) points before the occurrence of the diabetic neuropathy ("t0").

The occurrence relation was incidence of Diabetic neuropathy = f (Comorbidities, glycemic control, alcohol drinking+ physical activity + hypertension+ type of treatment+ WBC count + RBC count).

Domain: All type-1 and type-2 DM patients at risk for developing diabetic neuropathy.

## Nomogram and classification and regression tree development

Nomogram was developed to predict [52] individual risk of diabetic neuropathy among DM patients at diabetic clinic of university of Gondar comprehensive specialized hospital and Felege Hiwot referral hospital. The total nomogram score were classified as low risk, intermediate and high-risk scores to see size of each category. Classification and Regression Tree (CRT) analysis was performed to determine the complex interactions among the most potent candidate predictors in the final reduced model to build the classification trees. Youden-index was used to estimate cut off point for predictive probability of diabetic neuropathy at different threshold. By choosing the cutoff value get from Youden's index the sensitivity, specificity, NPV, PPV and accuracy was calculated and cut off value was selected depend availability of resources and aim of implemented program.

## Risk prediction model performance

The reduced multivariable model was evaluated by calibration and discrimination [53].

Model discrimination is the model's ability to discriminate between subjects with and without the diabetic neuropathy was estimated through the areas under the ROC curve. The area under the receiver operating characteristic curve (AUC) ranges between 0.5–1. Model calibration is examined the level of agreement between the diabetic neuropathy probabilities estimated by the model versus the observed diabetic neuropathy frequencies through graphical comparison, calibration plot. Good calibration means that the estimated probability of diabetic neuropathy is similar to the observed diabetic neuropathy frequency. Statistically we checked by the Homer and Lemeshow test with a significant P- value implies that the model is not well calibrated as it performs differently for different risk categories. Finally the results were presented using statement, table, figures and reported according to transparent reporting of multivariable prediction model for individual progress or diagnosis (TRPOID)statement [54].

## Internal validation

In this study by using bootstrapping method internal validation was done to avoid over fitting of the model [53]. The model was developed in the full sample. Then multiple random minimum samples (1000) draw from the full sample. The calibration and discrimination of each bootstrap model were compared to the corresponding estimates of the bootstrap models when applied in the original full sample. These differences were averaged, and provide an indication of the average optimism of the bootstrap models.

## Decision curve analysis

Decision curve analysis was done to evaluate prediction models that is important for medical decision making to overcome the limitations discrimination and calibration [55]. It was done graphically for model, treatment for all, and treatment for none. The net benefit of the developed model was done by decision curve plot.

**Mobile based application of Nomogram.**   To make the nomogram accessible to physicians and patients, mobile based application tool were developed for predicting risk of diabetic neuropathy in diabetes mellitus patients. The mobile based tool was developed by "sublime text" application with java script programming.

# Results

The flow chart of participates selection was displayed (Fig 1).

## Socio-demographic and behavioral characteristics of DM patients

A total of 808 diabetes mellitus patients who had follow up visit were included. The mean baseline age of diabetes mellitus patients was 45.6±3.1 years. More than three-fourth 616(76.2%) diabetes mellitus patients were from urban residents (Table 1).

## Diagnosis and treatment related characteristics of DM patients

More than two-third of patients 545(67.5%) were type-2 DM patients and half of them 401 (49.6%) diabetes mellitus patients were used oral drugs medication. Nearly half of the diabetes mellitus patients 385(47.6%) had other comorbidities. Majority of diabetes mellitus patients 557(71.4%) had good adherence (Table 2).

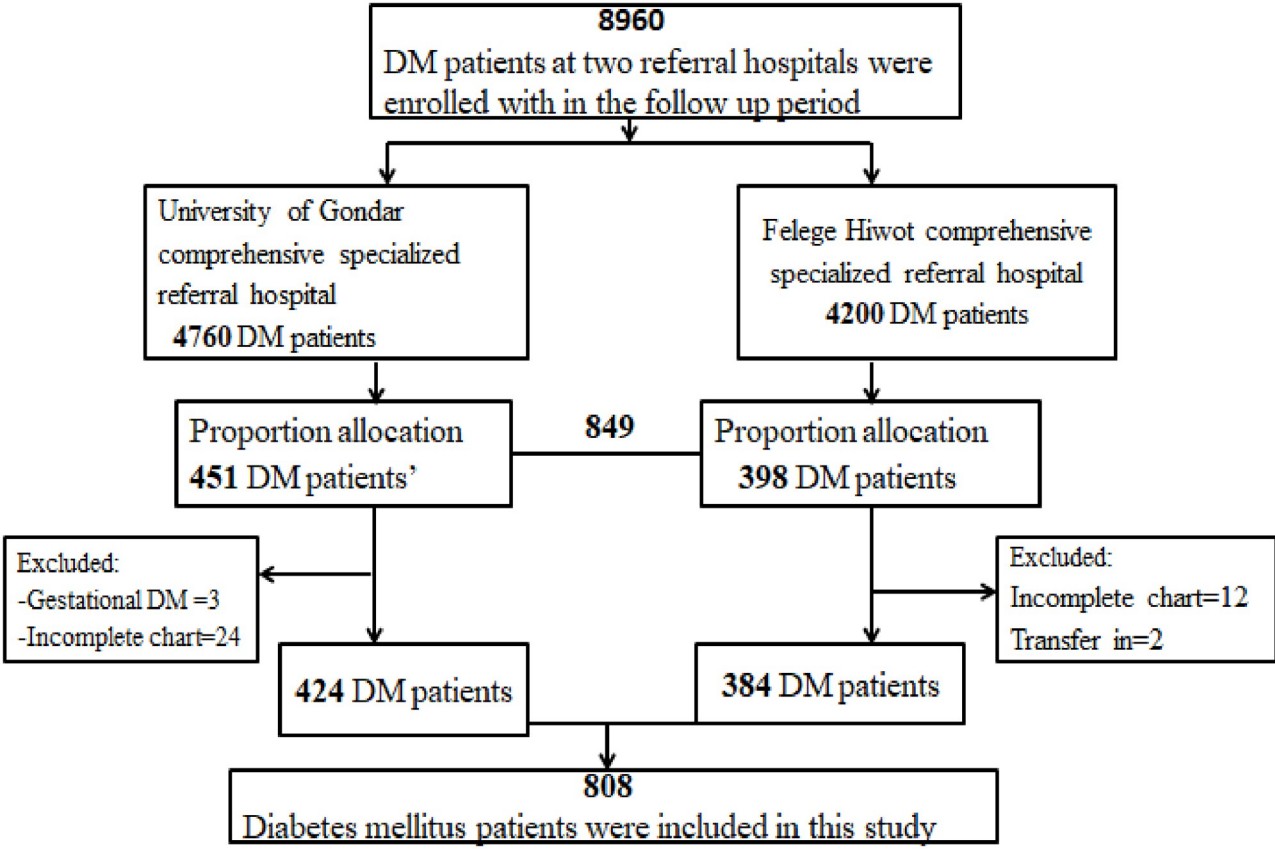

**Fig 1. Flowchart of participant selection for the development of nomogram to develop and validation of risk prediction model for diabetic neuropathy among diabetes mellitus patients at selected referral hospitals, in Amhara regional state Northwest Ethiopia, 2005–2021.**

### Laboratory related characteristics of diabetes mellitus patients

The mean value of fasting blood glucose level was 92 ± 15 and the mean total cholesterol and total triglyceride level had 189.9 ± 58.7 and 170.5 ± 73.7 mg/dl respectively (Table 3).

### Incidence of diabetic neuropathy

The incidence proportion of diabetic neuropathy was 172 (21.29%, 95% CI; 18.59, 24.25) Type-2 diabetes mellitus 137(79.65%) had highest incidence of diabetic neuropathy whereas 35(20.35%) were found in type-1 diabetes mellitus patients (Fig 2).

### Prediction model development and validation

### Selected predictors

In this risk prediction model socio-demographic, diagnosis and treatment related, laboratory and behavioral related predictors were considered for development of prediction model of diabetic neuropathy. The candidate predictors were selected by least absolute shrinkage and selection operator (LASSO) regression for the development and validation of model for diabetic neuropathy (Table 4). Multi- collinerity test was checked by VIF and unhealthy diet was not included in our model due to high VIF (17.8).

**Table 1. Socio-demographic and behavioral predictors of for develop and validation of risk prediction model for diabetic neuropathy among diabetes mellitus patients at selected referral hospitals, in Amhara regional state Northwest Ethiopia, 2005–2021 (n = 808).**

| Predictors | Diabetic Neuropathy | | Frequency | Percentage |
|---|---|---|---|---|
| | Yes | No | | |
| Sex | | | | |
| Male | 73 | 327 | 400 | 49.5 |
| Female | 99 | 309 | 408 | 50.5 |
| Baseline age | | | | |
| < 45 years | 62 | 321 | 383 | 47.4 |
| > = 45 years | 110 | 315 | 425 | 52.6 |
| Residence | | | | |
| Rural | 26 | 166 | 192 | 23.8 |
| Urban | 146 | 470 | 616 | 76.2 |
| Alcohol drinking | | | | |
| Yes | 74 | 98 | 172 | 21.3 |
| No | 287 | 349 | 636 | 78.7 |
| Physical activity | | | | |
| Yes | 311 | 325 | 636 | 78.7 |
| No | 75 | 97 | 172 | 21.3 |
| Unhealthy diet | | | | |
| Yes | 83 | 89 | 172 | 21.3 |
| No | 321 | 315 | 636 | 78.7 |

Thirteen variables were selected by Lasso regression and included in the multiple logistic regression models (Table 5). Finally, we construct the clinical risk prediction model with eight statistically significant predictors including hypertension, glycemic control (FBG), other comorbidities, Alcohol drinking, physical activity, type of treatment for DM, WBC and RBC count. Goodness of fit test was checked had insignificant p-value 0.3077, best fitted model.

## Prediction model development using original beta coefficients

By using area under ROC curve the discriminating power of the model was evaluated and individual predictors in the final reduced model had poor performance discriminating the risk of diabetic neuropathy among DM patients starting from 51.1% to 62.4% but they had good discriminating ability in combined effect. The predictive performance of model using combination of other comorbidities, glycemic control, physical activity and WBC had 70.0%. The predictive performance powers of each predictor were displayed (S2 File). The area under curve of the final reduced model using original beta coefficients were 73.49% (95% CI; 69.3%, 77.6%) (Fig 3A). The developed model had calibration test value (p-value = 0.451) well calibrated model, the model well represented the data that was agreement between observed and the predicted probability (Fig 3B).

The probability for risk of diabetic neuropathy using original beta coefficients was

$$P(\text{Diabetic-neuropathy}) = 1/1 + \exp - (-1.96 + 0.9*\text{RBC(low)} - 1.14*\text{WBC(low)}$$

$$+0.4*\text{Hypertension (yes)} - 1.79*\text{Physical activity (yes)}$$

$$+1.13*\text{Alcohol drinking (yes)} + 1.29*\text{other comorbidities (yes)}$$

$$+0.58*\text{Glycemic control (poor)} - 0.72*\text{type of treatment (insulin only)}$$

**Table 2. Clinical predictors for develop and validation of risk prediction model for diabetic neuropathy among diabetes mellitus patients at selected referral hospitals, in Amhara regional state Northwest Ethiopia, 2005–2021 (n = 808).**

| Predictors | Diabetic Neuropathy | | Frequency | Percentage |
|---|---|---|---|---|
| | Yes | No | | |
| Type of diabetes mellitus | | | | |
| Type-1 DM | 35 | 228 | 263 | 32.5 |
| Type-2 DM | 137 | 408 | 545 | 67.5 |
| Type of treatment for diabetes mellitus | | | | |
| Insulin only | 55 | 293 | 348 | 43.1 |
| Oral drugs only | 96 | 305 | 401 | 49.6 |
| Insulin and oral drugs | 21 | 38 | 59 | 7.3 |
| Body mass index in kg/m$^2$ | | | | |
| <18.5 | 9 | 28 | 37 | 4.6 |
| 18.5–24.9 | 54 | 191 | 245 | 30.3 |
| > = 25 | 109 | 417 | 526 | 65.1 |
| Mean arterial blood pressure | | | | |
| >100 mmHg | 62 | 152 | 214 | 26.5 |
| 70–100 mmHg | 110 | 484 | 594 | 73.5 |
| Other comorbidities | | | | |
| Yes | 50 | 373 | 423 | 52.4 |
| No | 122 | 263 | 385 | 47.6 |
| Adherence | | | | |
| Poor | 52 | 179 | 231 | 28.6 |
| Good | 120 | 457 | 577 | 71.4 |
| Hypertension | | | | |
| Yes | 99 | 244 | 343 | 42.5 |
| No | 73 | 392 | 465 | 57.5 |
| Duration of DM in years | | | | |
| <6 years | 71 | 257 | 328 | 40.6 |
| 6–11 years | 79 | 316 | 395 | 48.9 |
| >11 years | 22 | 63 | 85 | 10.5 |

**Table 3. Laboratory related predictors for develop and validation of risk prediction model for diabetic neuropathy among diabetes mellitus patients at selected referral hospitals, in Amhara regional state Northwest Ethiopia, 2005–2021 (n = 808).**

| Predictor | Mean ± SD |
|---|---|
| Mean fasting blood glucose(mg/dl) | 92 ± 15 |
| Hemoglobin(g/dl) | 13.9 ± 1.8 |
| Total cholesterol level(mg/dl) | 189.9 ± 58.7 |
| Total triglyceride level(mg/dl) | 170.5 ± 73.7 |
| Creatinine(mg/dl) | 0.9 ± 0.6 |
| White blood cells(Cells/microliter) | 6.9 ± 2.8 x10$^3$ cells/mcl |
| Red blood cells(Cells/microliter) | 4.8 ± 0.9x10$^6$ cells/mcl |
| Platelets(Cells/microliter) | 233 ± 74x10$^3$ cells/mcl |

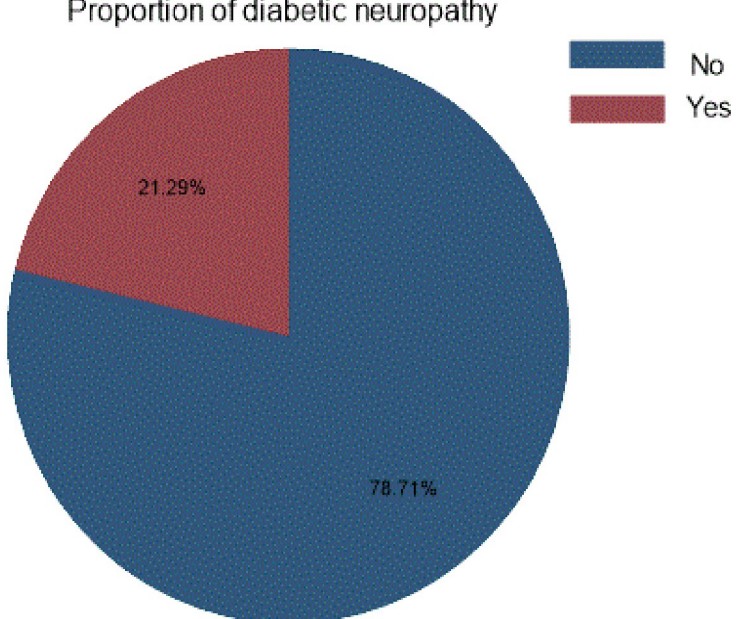

**Fig 2. Pie chart for the proportion diabetic neuropathy to develop and validation of risk prediction model for diabetic neuropathy among diabetes mellitus patients at selected referral hospitals, in Amhara regional state Northwest Ethiopia 2005–2021.**

## Prediction model development using nomogram

For the sake of simplicity and easy to use in clinical practice the model was developed by nomogram (Logistic regression). All significant regression coefficients in the final reduced model were used for model development. The model presented by monogram through eight significant variables that predicts diabetic neuropathy was displayed (Fig 4). From nomogram the possible minimum and maximum score was 0 to 47.2 with different threshold probabilities to develop diabetic neuropathy. The cut off value to nomogram score as the probability of 0.24 was 26. Hence, DM patients who had <26 score from nomogram total score was low risk whereas those who had > = 26 score from total score was as high risk to develop diabetic neuropathy.

The performance of developed nomogram in predicting diabetic neuropathy was AUC = 73.2% (95% CI; 69.0%, 77.3%), had good level of accuracy (Fig 5A) and calibration test (p-value 0.502), the model was well calibrated model, indicates the observed probability was agreement with expected probability (Fig 5B).

**Table 4. Optimum shrinkage factor (lambda) and potential predictors identified by lasso regression by 10-fold cross validation selection method for develop and validation of risk prediction model for diabetic neuropathy among diabetes mellitus patients at selected referral hospitals, in Amhara regional state Northwest Ethiopia 2005–2021 (n = 808).**

| ID | Description | Lambda | No. of non- zero coefficients | 0ut of sample dev. Ratio | CV mean deviance |
|---|---|---|---|---|---|
| 1 | First lambda | 0.09923 | 0 | 0.0023 | 1.037898 |
| 21 | Lambda before | 0.015437 | 12 | 0.0575 | 0.9759118 |
| *22 | **selected lambda** | **0.0140656** | **13** | **0.0578** | **0.0756381** |
| 23 | Lambda after | 0.012816 | 13 | 0.0576 | 0.9757846 |
| 27 | Last lambda | 0.0088336 | 19 | 0.0553 | 0.9782416 |

**Table 5. Multi variable logistic regression for variables retained in the final reduced model for develop and validation of risk prediction model for diabetic neuropathy among diabetes mellitus patients at selected referral hospitals, in Amhara regional state Northwest Ethiopia, 2005–2021 (n = 808).**

| Predictors | Diabetic neuropathy | | Multi variable analysis | | Model reduction |
|---|---|---|---|---|---|
| | No | Yes | Beta coefficients (95% CI) | P-value | P- value |
| Baseline age in years | | | | | |
| <45 years | 321 | 62 | 0 | | |
| > = 45 years | 315 | 110 | 0.15(-0.28,0.59) | 0.516 | |
| Sex | | | | | |
| Female | 309 | 99 | 0 | | |
| Male | 327 | 73 | -0.26(-0.63,0.10) | 0.157 | |
| Residence | | | | | |
| Rural | 166 | 26 | 0 | | |
| Urban | 470 | 146 | 0.28(-0.23,0.79) | 0.279 | |
| Type of DM | | | | | |
| Type-1 DM | 228 | 35 | 0 | | |
| Type-2 DM | 408 | 137 | 0.024(-0.6, 0.7) | 0.944 | |
| Mean arterial blood pressure | | | | | |
| 70–100 mmHg | 484 | 110 | 0 | | |
| >100 mmHg | 152 | 62 | 0.15(-0.25,0.56) | 0.459 | |
| Type of treatment | | | | | |
| Both insulin and oral drug | 38 | 21 | 0 | | |
| Insulin only | 305 | 96 | -0.49(-1.32,0.10) | 0.05 | 0.02 * |
| Oral drugs only | 293 | 55 | -0.61(-1.09,0.15) | 0.139 | |
| Baseline glycemic control | | | | | |
| <130mg/dl | 148 | 25 | 0 | | |
| > = 130mg/dl | 488 | 147 | 0.58(-0.01, 1.16) | 0.04 | 0.04* |
| Other comorbidities | | | | | |
| No | 373 | 50 | 0 | | |
| Yes | 263 | 122 | 1.29(0.85,1.68) | 0.000 | 0.00* |
| Alcohol drinking | | | | | |
| No | 349 | 287 | 0 | | |
| Yes | 98 | 74 | 1.13(.015, 2.25) | 0.04 | 0.003* |
| Physical Activity | | | | | |
| No | 97 | 75 | 0 | | |
| Yes | 325 | 311 | -1.79(-2.94,-0.7) | 0.002 | 0.001* |
| Hypertension | | | | | |
| No | 392 | 73 | 0 | | |
| Yes | 244 | 99 | 0.412 (0.00, 0.82 | 0.04 | 0.01* |
| White blood cells/microliter | | | | | |
| 4500–11000 | 266 | 82 | | | |
| <4500 | 33 | 20 | -1.14(-2.2, -0.01) | 0.04 | 0.03* |
| >11000 | 337 | 70 | 0.15(-0.58, 0.89) | 0.684 | |
| Red blood cells/microliter | | | | | |
| 4–6 million | 270 | 80 | 0 | | |
| <4 million | 34 | 20 | 0.90(.056, 1.74) | 0.03 | 0.03* |
| >6 million | 332 | 72 | -0.24(-0.99 0.52) | 0.539 | |
| _cons | | | -1.96(-2.97,0.94) | 0.000 | |

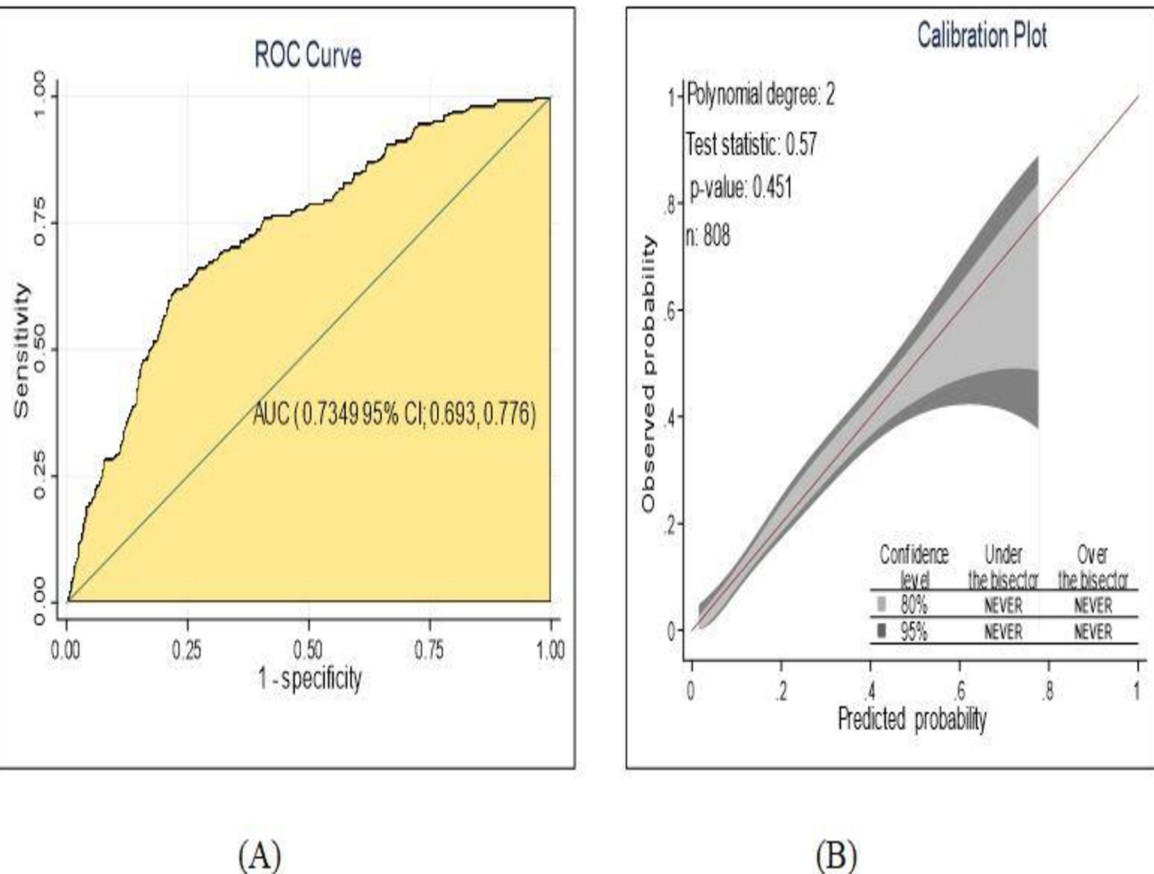

**Fig 3. The performance of the model using original beta coefficients A) AUC B) calibration plot, for develop and validation of risk prediction model for diabetic neuropathy among diabetes mellitus patients at selected referral hospitals, in Amhara regional state Northwest Ethiopia 2005–2021.**

### Prediction model development using machine learning by classification and regression tree

Besides the model developed by nomogram, it also presented by machine learning algorism with classification and regression tree analysis suitable when continuous and categorical variables used that predict diabetic neuropathy. The prediction rate (accuracy) for diabetic neuropathy in classification and regression tree was 78.7% (Table 6).

From eight significant variables used for nomogram development, the classification and regression tree (CRT) select five most potent predictors includes other comorbidities, glycemic control, physical activity, White blood cells and red blood cells count. Presence of comorbidities was the most important node that predicts diabetic neuropathy, followed by glycemic control and physical activity; low RBC and high WBC count. For instance, from 172 having diabetic neuropathy cases, 59(43.1%) had presence other comorbidities and no doing physical activity were develop diabetic neuropathy. In classification and regression tree there were four depths, six terminal nodes used for decisions (Fig 6).

The performance of the classification and regression decision tree was good accuracy with AUC = 70.2% (95% CI; 65.8%, 74.6%) (Fig 7A) and well calibrated model (p-value -0.412) (Fig 7B), which means the predicting and observed probability was the same.

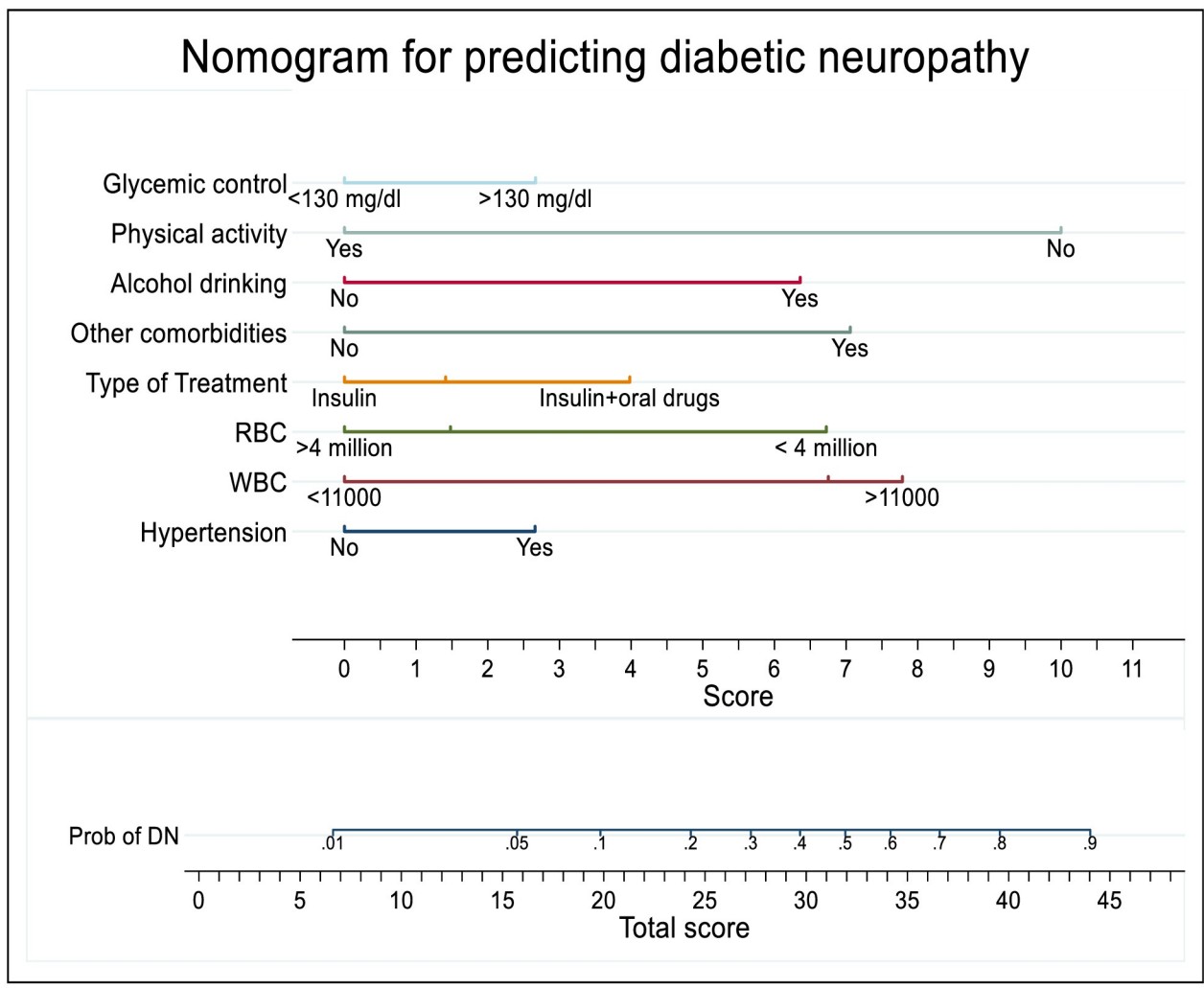

**Fig 4. Nomogram to develop and validation of risk prediction model for diabetic neuropathy among diabetes mellitus patients at selected referral hospitals, in Amhara regional state Northwest Ethiopia, 2005–2021.**

### Internal validation

The area under ROC curve was assessed based on the bootstrap dataset coefficients. The developed nomogram was internally validated had AUC; 71.7(95% CI; 67.3%, 76.0%) (Fig 8A) and well calibration model (p-value- 0.945) (Fig 8B).

After internal validation the discriminating performance of nomogram was comparable with less optimism coefficients found to be 0.015, which indicates less likely over fitting model and bias. The average calculated value of optimism to the performance model was 0.0075.

The performance of the classification and regression decision tree after bootstrapping was the same as the developed CRT decision tree, good accuracy with AUC = 70.2% (95% CI; 65.8%, 74.6%) (Fig 9A) and well calibrated model (p-value:-0.389) (Fig 9B). The optimism coefficient was 0.000.

In both nomogram and classification and regression tree (CRT) the discrimination and calibration performance power were comparable, good accuracy so that, they were used to predict the risk of diabetic neuropathy among DM patients (Fig 10).

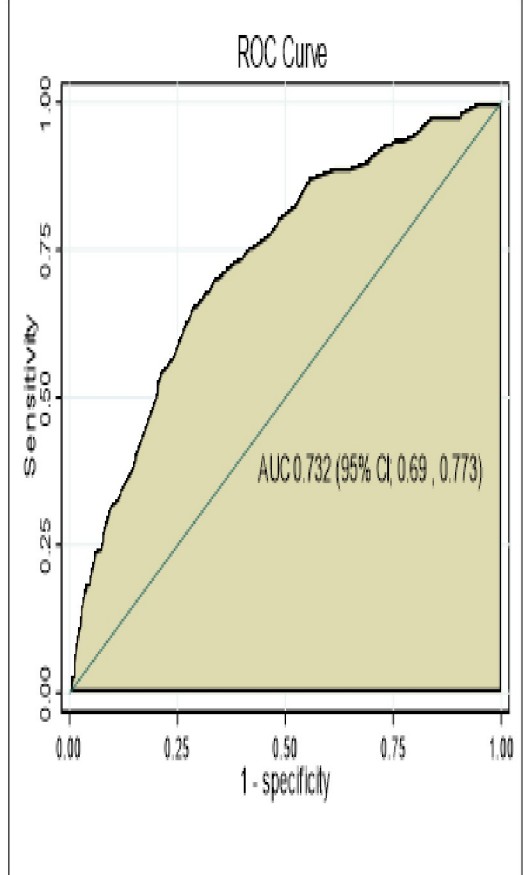
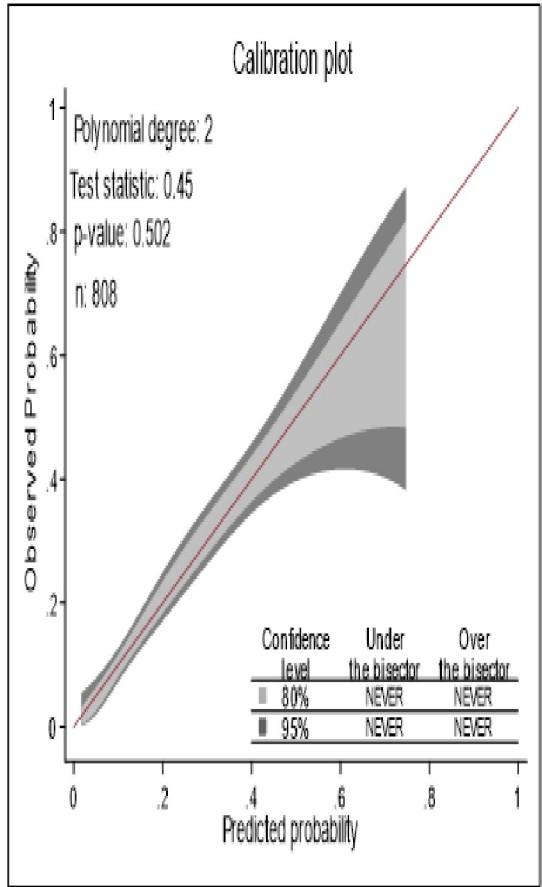

(A)                                                            (B)

**Fig 5. Performance of model using nomogram A) AUC B) Calibration plot, prediction for develop and validation of risk prediction model for diabetic neuropathy among diabetes mellitus patients at selected referral hospitals, in Amhara regional state Northwest Ethiopia 2005–2021.**

## Cutoff point for probability of diabetic neuropathy by nomogram and CRT

Using the nomogram, optimal cutoff point for predicted probability for risk of diabetic neuropathy by using Youden's index(J) was 0.2447 and the sensitivity, specificity, positive predictive value and negative predictive value was 65.2%(95% CI; 58;0%, 72.2%), 71.7%(95% CI;

**Table 6. Prediction rate for develop and validation of risk prediction model for diabetic neuropathy among diabetes mellitus patients at selected referral hospitals, in Amhara regional state Northwest Ethiopia, 2005–2021.**

| Classification | | | |
|---|---|---|---|
| Observed | Predicted | | |
| | No | Yes | Percent Correct |
| No | 636 | 0 | 100.0% |
| Yes | 172 | 0 | 0.0% |
| Overall Percentage | 100.0% | 0.0% | 78.7% |

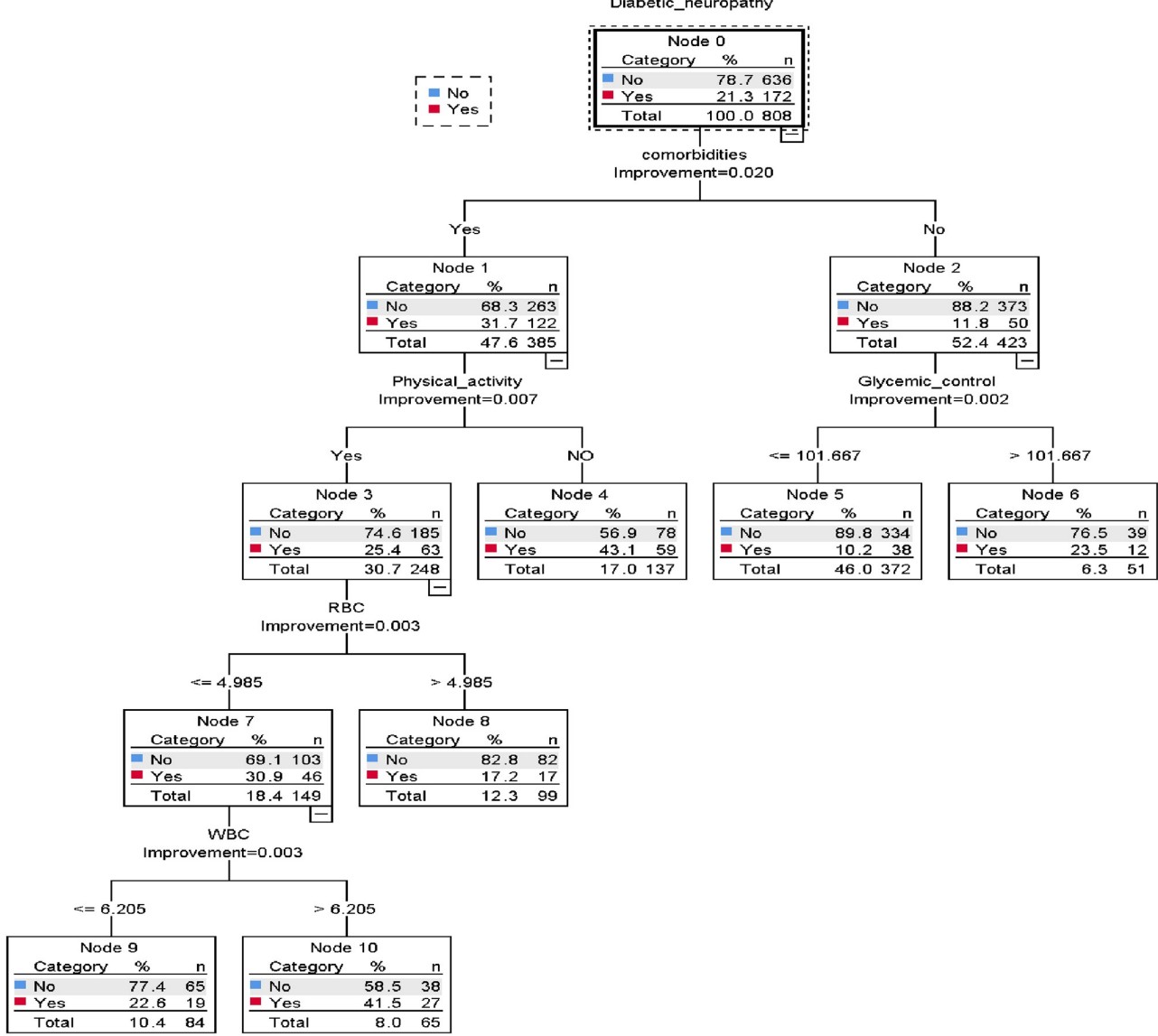

**Fig 6. Classification and regression decision tree for develop and validation of risk prediction model for diabetic neuropathy among diabetes mellitus patients at selected referral hospitals, in Amhara regional state Northwest Ethiopia 2005–2021.**

68.0%, 75.2%), 38.4% (95% CI; 32.7%, 44.2%), 88.4% (95% CI; (85.3%, 91.0%) respectively (Table 7).

In classification and regression tree(CRT) using optimal cutoff point for predicting the probability risk of diabetic neuropathy by using Youden's index was 0.1847 and the sensitivity, specificity, positive predictive value and negative predictive value was 72.09 (95% CI; 64.8%, 78.6%), 57.7%(95% CI; 53.8%, 61.6%), 31.5% (95% CI; 27.0%, 36.4%), 88.4% (95% CI; (84.9%, 91.3%) respectively.

## Risk classification for diabetic neuropathy using nomogram

For easily practical utility we developed nomogram by using regression coefficients in the final reduced model. The proportion of diabetes mellitus patients to risk of diabetic neuropathy at

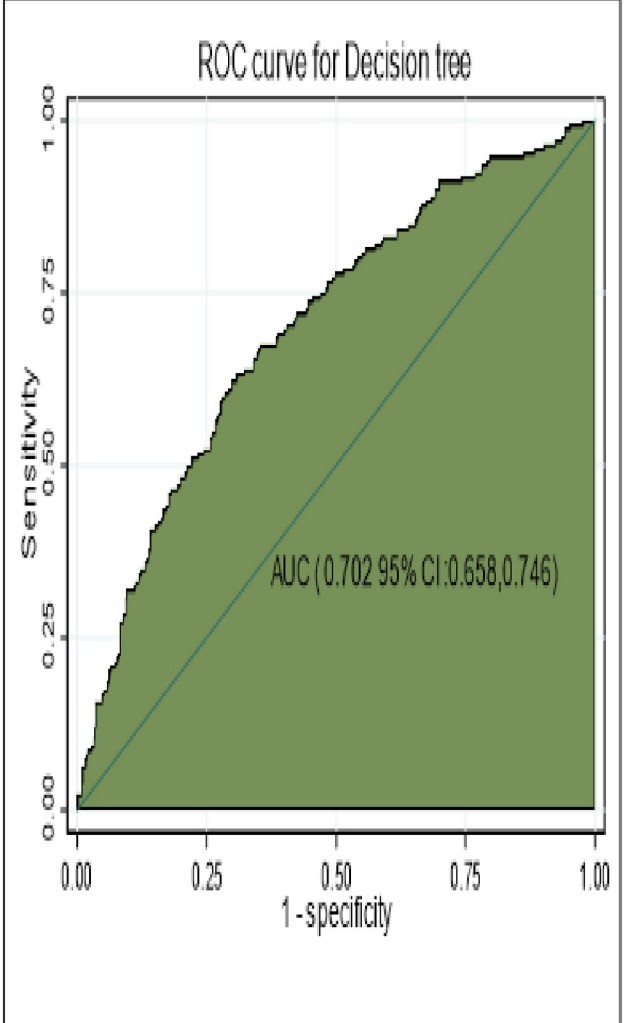
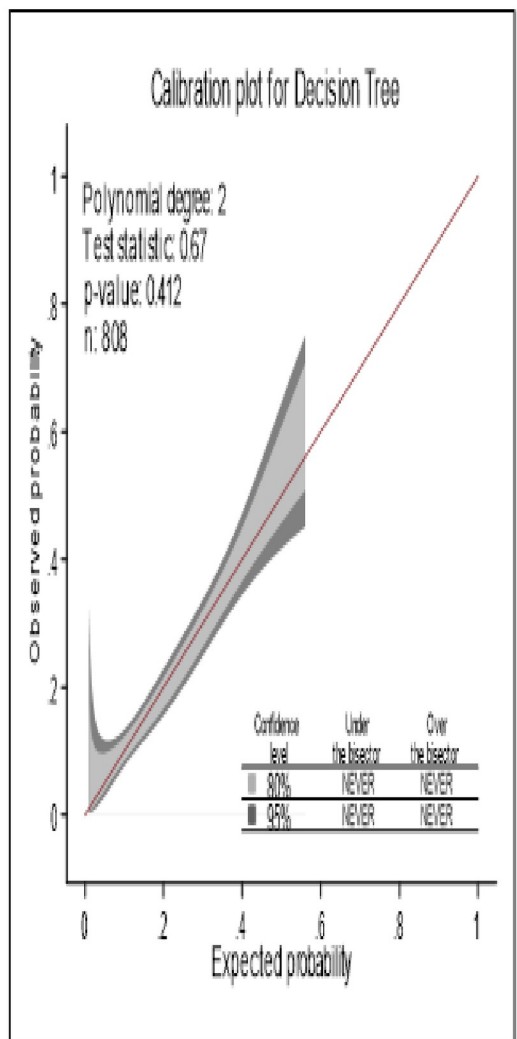

(A)                    (B)

**Fig 7. The performance of the model using decision tree A) AUC B) calibration plot to develop and validation of risk prediction model for diabetic neuropathy among diabetes mellitus patients at selected referral hospitals, in Amhara regional state Northwest Ethiopia 2005–2021.**

low risk was 516(63.8%), intermediate risk 282(34.90%) and high risk group 10(1.24%). When the risk classified as two categories the proportions of DM patients for risk of diabetic neuropathy at low risk was 516(63.86%) and high risk 292(36.14%) (Table 8).

In classification and regression tree (CRT), the proportion diabetes mellitus patients to the risk of diabetic neuropathy at low risk was 150(18.56%), intermediate risk 636(78.72) and high risk 22(2.72%).

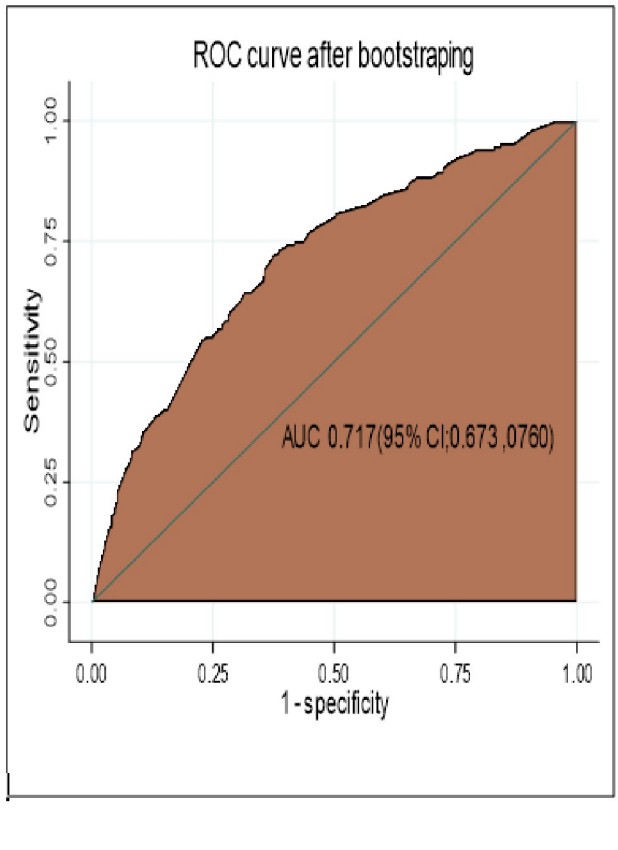
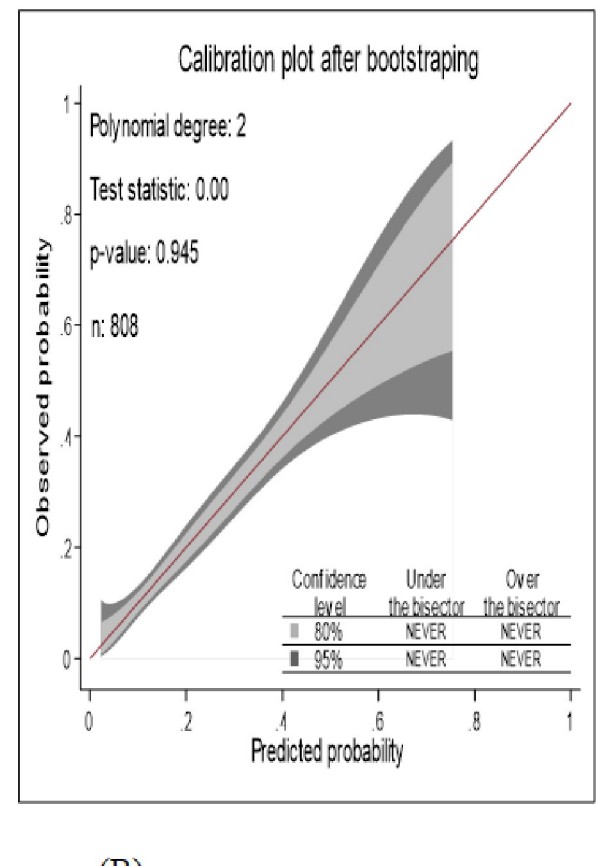

(A)                                          (B)

**Fig 8. The performance of the model using nomogram after bootstrapping A) Area under the curve (AUC) B) calibration plot, for develop and validation of risk prediction model for diabetic neuropathy among diabetes mellitus patients at selected referral hospitals, in Amhara regional state Northwest Ethiopia, 2005–2021.**

## Decision curve analysis

Besides model performance assessed by AUC, calibration plot, clinical and public health utility of the model was also assessed by decision curve analysis. The developed model using nomogram and classification and regression tree (thick red line} had highest net benefit from threshold probabilities greater than 0.1(10%) compared treatments none (thick black line), had high cost benefit ratio. The model has no or similar net benefit to treatment all (thin black line) regardless of their risk across thresholds below 0.1 (10%) (Fig 11A and 11B).

**Mobile-based application of nomogram.**   Where physicians' or diabetes mellitus patients enter their selected clinical and laboratory data to the developed mobile based tool, then the tool was giving alert as cutoff value 26 get from nomogram, weather they are at low risk (<26) or high risk (> = 26) of diabetic neuropathy by calculating cumulating score in order to keep or improve their health condition. When we enter data to the tool had score 21 (Fig 12).

## Discussion

This risk prediction model used a multicenter retrospective follow up study to develop a practical tool for prediction of diabetic neuropathy among both type-1 and type-2 DM patients. Our study established a diabetic neuropathy risk prediction model using combined set

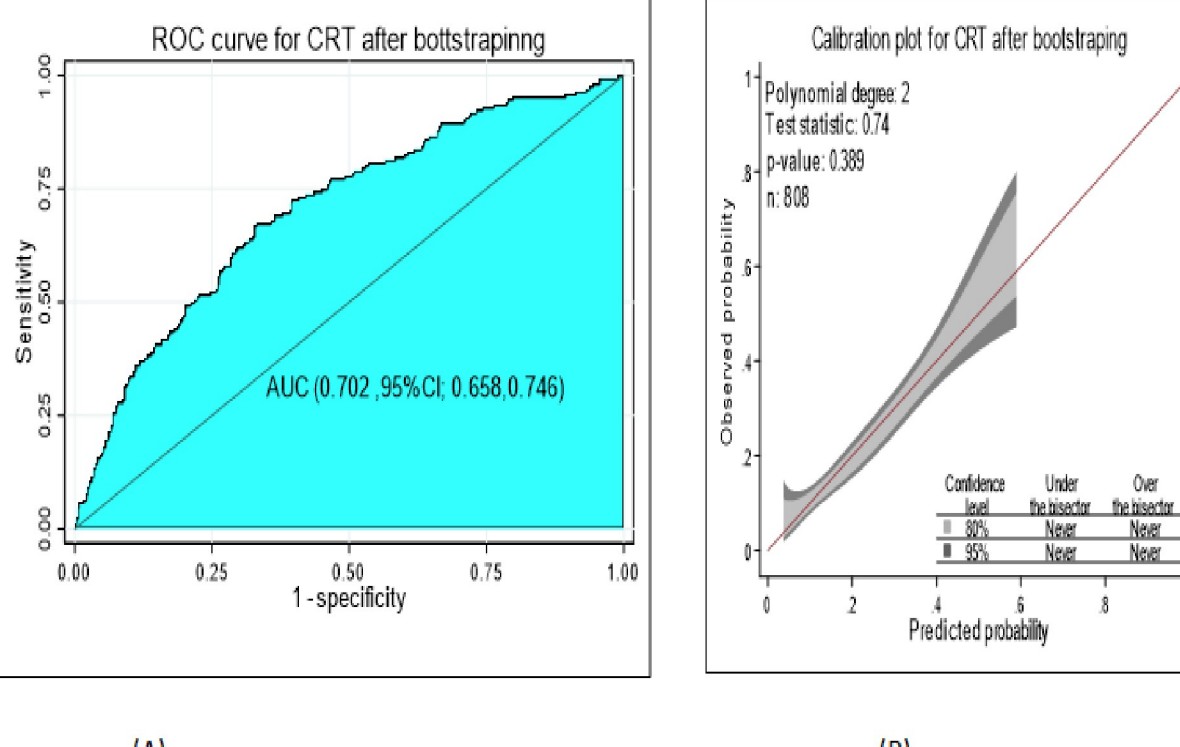

**Fig 9. The performance of the model using classification and regression decision tree after bootstrapping A) AUC B) Calibration plot to develop and validation of risk prediction model for diabetic neuropathy among diabetes mellitus patients at selected referral hospitals, in Amhara regional state Northwest Ethiopia 2005–2021.**

predictors to be used in the primary care settings, by developing nomogram and decision tree, aid in public health and clinical decision making for clinicians and to patients.

There were challenges to diagnose diabetic neuropathy, which needs electro physical and autonomic tests by mono filaments, is not available in low level health care system [56, 57]. Hence, predicting the risk of diabetic neuropathy in diabetes mellitus patients using easily measurable predictors were essential to take appropriate measures accordingly.

The study finding showed that the incidence proportion of diabetic neuropathy was 21.29% (95%CI; 18.59, 24.25), which is consistent with SRMA study in North Africa, 21.9% [16], Ethiopia, 22% [49], Qatar 23.0% [58] but lower than Tikur Anbesa and St. Paulous hospital,48.2% [18], SRMA in Africa 46% [13], Srilanka 28.8% [12].

These disparities might be number of participates involved with sample size difference, in case small sample size in this study [59], difference way of life style [60] and different level of health care system used for diagnosis. For instance, monofilament examination was not performed in our study area, results asymptomatic patients may not detect.

The model was developed by reducing twenty-two candidate predictors to thirteen potential predictors, which were passed to multivariable analysis. The model comprised eight variables identified as independent predictors of diabetic neuropath including glycemic control (FBG), other comorbidities, alcohol drinking, Hypertension, type of treatment for DM, physical activity, WBC and RBC count. This result is supported by previous studies conducted [36, 41, 46, 61–63].

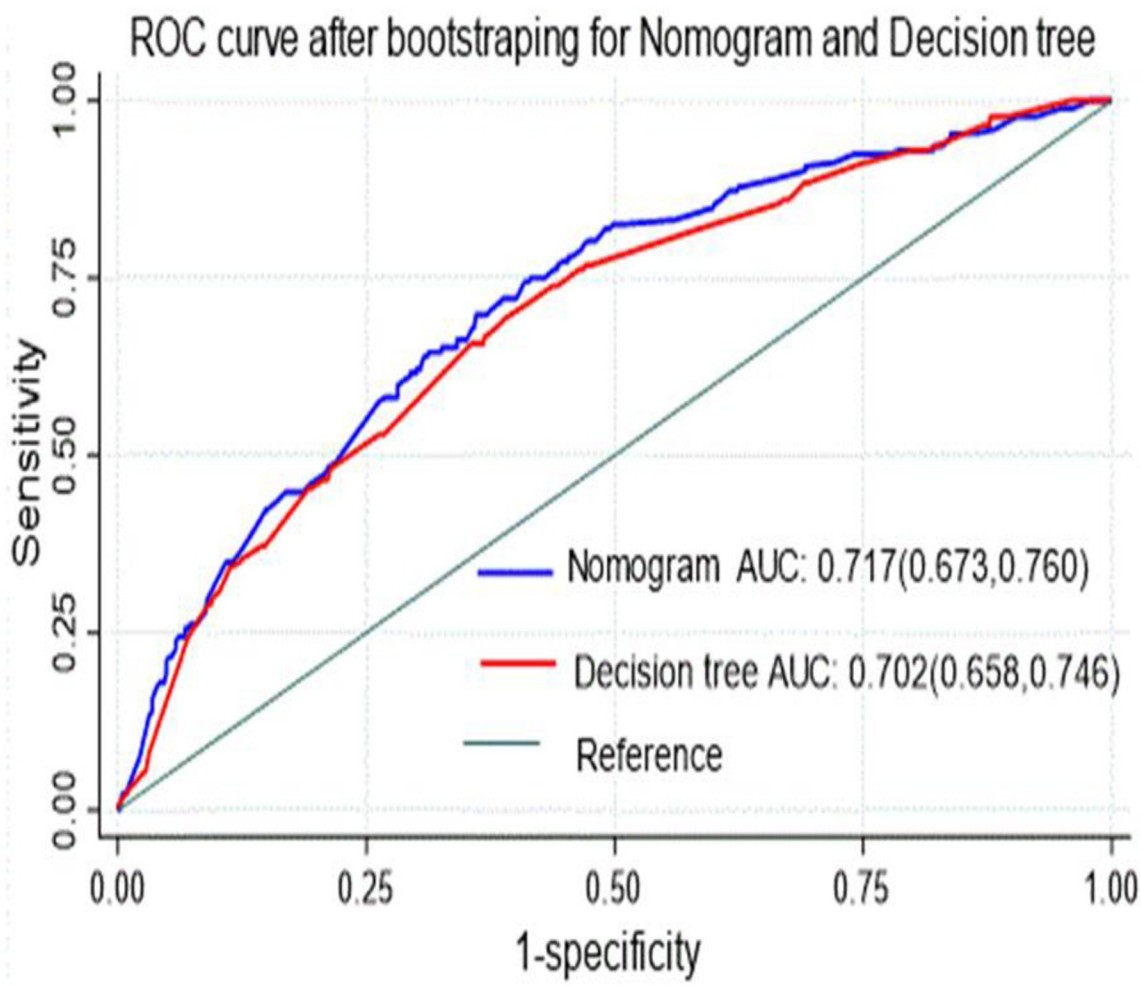

**Fig 10. Comparison of nomogram and decision tree for develop and validation of risk prediction model for diabetic neuropathy among diabetes mellitus patients at selected referral hospitals, in Amhara regional state Northwest Ethiopia 2005–2021.**

The prediction model developed from eight predictors and nomogram was developed found to have discrimination accuracy of AUC; 73.2% This result showed that a good accuracy discrimination power according to diagnostic accuracy classification criteria [64]. The model calibration assessed by calibration plot and Hosmer and Lemeshow test had p-value; 0.502, which had good agreement between observed and expected probability.

**Table 7. Sensitivity, specificity, PPV, NPV, LR+, LR- and accuracy of the nomogram at different cut-off points for develop and validation of risk prediction model for diabetic neuropathy among diabetes mellitus patients at selected referral hospitals, in Amhara regional state Northwest Ethiopia 2005–2021.**

| Cutoff | Sensitivity | Specificity | Accuracy | PPV | NPV | LR+ | LR_ |
|---|---|---|---|---|---|---|---|
| > = 0.1 | 72.1% | 63.5 | 65.4% | 35.3% | 90.4% | 1.9 | 0.4 |
| > = 0.2 | 70.4% | 66.4 | 67.2% | 36.1% | 89.2 | 2.1 | 0.4 |
| **> = 0.24** | **65.2%** | **71.7%** | **70.3%** | **38.4%** | **88.4** | **2.3** | **0.4** |
| > = 0.3 | 54.7% | 78.6 | 73.5% | 41.0% | 86.5% | 2.5 | 0.5 |
| > = 0.4 | 34.3% | 88.4 | 77.0% | 44.4% | 83.3% | 2.9 | 0.7 |

**Table 8. Risk classification for develop and validation of risk prediction model for diabetic neuropathy among diabetes mellitus patients at selected referral hospitals, in Amhara regional state Northwest Ethiopia, 2005–2021 (n = 808).**

| Risk categories' | Total number of DM patients | Incidence of diabetic neuropathy |
|---|---|---|
| Low risk($<$0.2) | 516(63.86%) | 60(7.4%) |
| Intermediate risk(0.2–0.6) | 282(34.90) | 107(13.3%) |
| High risk($>$0.6) | 10(1.24%) | 5 (0.6%) |
| Total | 808(100%) | 172(21.3%) |

The predictive role of each predictors was assessed for the identification of individuals at greater risk of diabetic neuropathy and its capacity of score from nomogram for glycemic control (poor), physical activity (no), other comorbidities (yes), alcohol drinking (yes), use combined insulin and oral drugs, hypertension (yes), WBC (high) and RBC (low) was 2.6, 10, 7, 6.4, 4, 2.6, 7.8 and 6.8 respectively.

The model was internally validated using the bootstrapping method of 1000 repetitions with replacement. The discriminating performance of the model after internal validation had also good discrimination power of AUC; 71.7% with well calibrated model (p value = 0.945). The optimism coefficient was found to be 0.015(1.5%), ensures that the model is less likely over fitting and less sample dependent, can easily and accurately individualized prediction of the risk of diabetic neuropathy.

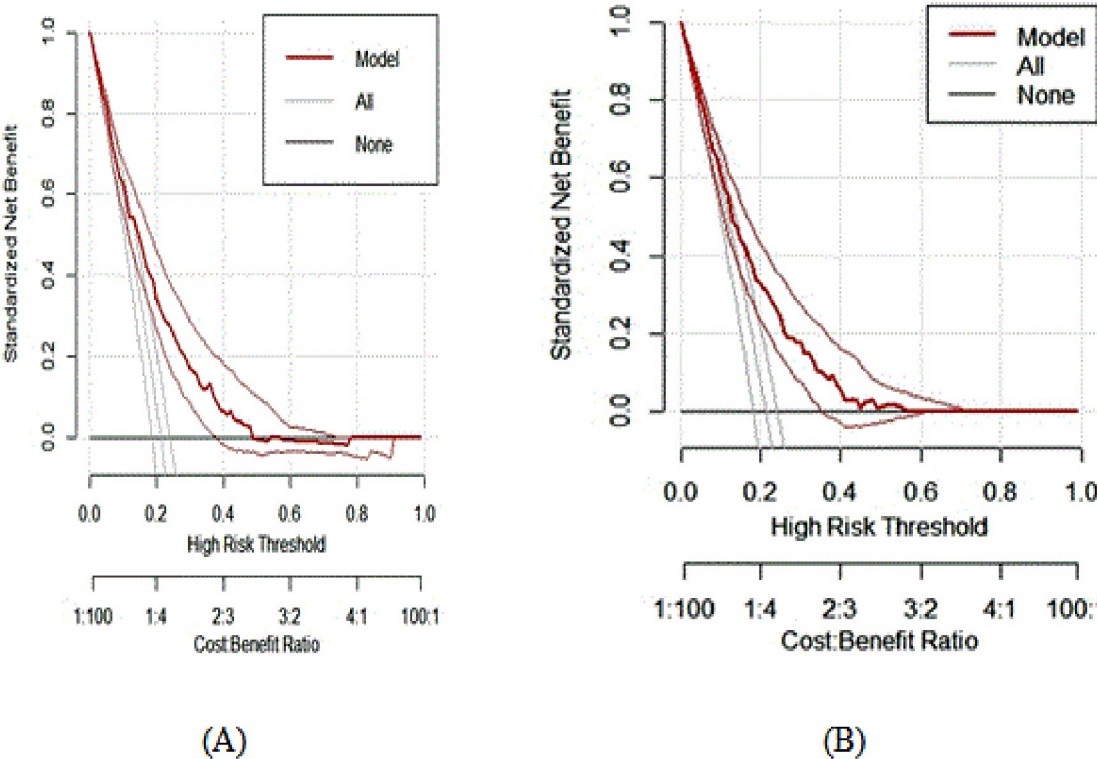

(A)                                                                    (B)

**Fig 11. Decision curve plot A) Nomogram B) Decision tree, showing the net benefit of the developed model for carrying out a certain intervention measure to develop and validation of risk prediction model for diabetic neuropathy among diabetes mellitus patients at selected referral hospitals, in Amhara regional state Northwest Ethiopia, 2005–2021.**

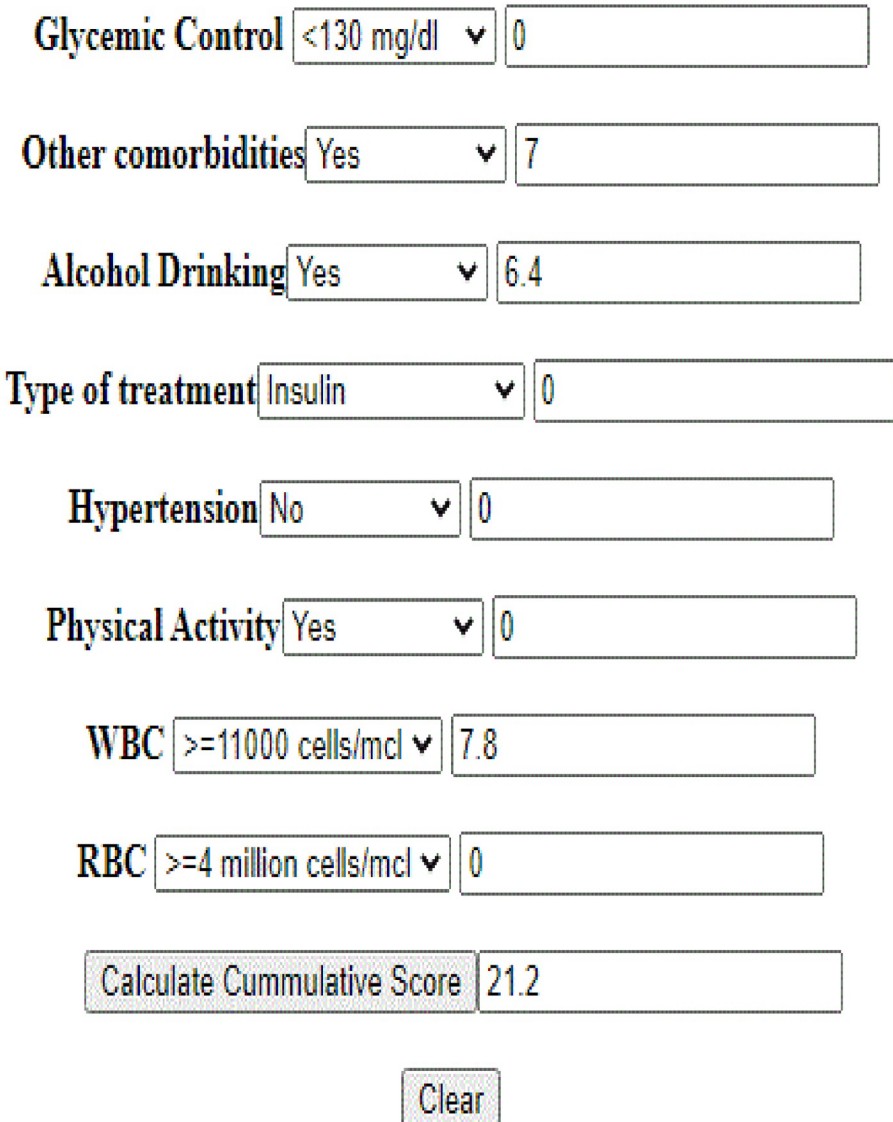

**Fig 12. Mobile based application outputs for develop and validation of risk prediction model for diabetic neuropathy among diabetes mellitus patients at selected referral hospitals, in Amhara regional state Northwest Ethiopia, 2005–2021.**

The performance of our model was found to be consistent with other prediction models developed to predict diabetic neuropathy among DM patients, using hypertension, age, heart rate and BMI as predictors, AUC; 71% in china [65] using hypertension, comorbidities, gender, age, obesity, abnormal triglycerides as predictors, AUC; 75% in china [66], but better than a study using glomerular filtration rate, glibenclamide and creatinine as predictors AUC = 66.05% in Mexico [67], using age, FBG, PBG, HbA1c, LDL, HDL and BMI as predictors AUC; 55.6% in china [68], using FBG, BMI, age as predictors AUC; 63.50 in Korea [69].

This might be due to difference in study participants involved through socio- demographic characteristics, difference number of predictors used in the model development [70]. The model calibration based on Hosmer and Lemshow test P-value; 0.945, is consistent another study 0.52 in china [68], indicates well calibrated model.

However, the discriminative performance of the model was lower than a study done in Italy using type of DM, smoke, BMI and HbA1c as predictors AUC; 76.9% [71] and AUC; 85.9% in china [39]. This might sample size variation, use variety of validation techniques, the lower number of predictors incorporated in case of our study compared to others, use machine learning algorism, to select the most potent predictors increases the power of the study [70].

In our nomogram prediction score, using 0.24 as cutoff point has an acceptable level of the sensitivity; specificity was 65.2%, 71.2%, respectively. Using 0.18 as cut off point in classification and regression tree (CRT) had also acceptable level of sensitivity; specificity was 72.09% and 57.7% respectively. Thus it was better to identify diabetic neuropathy cases and possible to shift the cutoff point to increase either of the accuracy measures depending on the aim of program and availability of resources.

The benefit that the developed nomogram and decision tree would add to clinical practice was also presented in the form of a decision curve analysis. In our study the decision curve analysis showed that there was high net benefit than using treat all or treat none strategies when the threshold probability of the patient greater than 0.1(10%) in both nomogram and classification and regression tree(CRT). However, the model is not useful for threshold probabilities below 10%. Thus, threshold probabilities are the most important components of the decision curve analysis depending on which a clinician can decide on whether to use the model or not when there is a need to carry out an intervention for patients at risk of diabetic neuropathy.

Although the autonomic tests by monofilament examination and nerve conduction studies gives better diagnosis of diabetic neuropathy [72]. However, prediction of diabetic neuropathy among type-1 and type-2 DM patients using clinical and behavioral characteristics of alone enabled to identify low and high risk diabetes mellitus patients.

This prediction model is not a replacement of electro physical diagnosis and monofilament examination of diabetes mellitus patients to diagnose diabetic neuropathy [73]; however, it will be as screening tool in resource poor settings for further diagnostic workup and management options. Besides the nomogram and decision tree is easier to use in routine clinical and public health practice than regression models and has comparable discrimination and calibration.

This study has following strengths. Firstly, it was conducted multicenter study and mobile based application toll was developed for easily applicability. Secondly, the model was internally validated using bootstrapping technique, resulted small optimism coefficient, indicating our model is less likely over fitted model, can easily predict the risk of diabetic neuropathy. Thirdly, this prediction model is constructed from easily obtainable clinical and behavioral predictors that make it applicable in primary care settings. Finally the model was also developed by classification and regression tree analysis to show the interaction effect of each predictor to the risk of diabetic neuropathy.

However, the findings from this study should be interpreted with the perspective of the following limitations. Even if multicenter studies, the model did not validate in separate datasets so that it needs external validation before using it in another context. In addition most diabetic neuropathy patient was asymptomatic may missed cases, who had no sign and symptoms. The study had many missing predictors due to retrospective study design. Generally the model will provide its maximum benefit in clinical practice provided that all the required predictor information is collected and developing mobile based application makes it the nomogram easily applicable for clinicians and to patients.

## Conclusion and recommendation

The model was developed from precisely measured, clinical and behavioral predictors found in primary health care setting includes glycemic control(FBG), hypertension, physical activity, other comorbidities, alcohol drinking, type of treatment for DM, WBC and RBC count. The model was presented by nomogram and decision tree with having good discriminating performance power and well calibrated. It was internally validated by bootstrapping techniques with small optimum coefficients, less likely of over fitting of the model. The nomogram was better than decision tree, helps us to do a risk stratification of diabetes mellitus patients and to identify those at higher risk of having diabetic neuropathy. Subsequently, high-risk groups can be linked to a center, which is equipped with electro physical diagnosis and monofilament examination centers for further assessment and better management. The nomogram and decision tree had added net benefit in clinical practice as it was assured by decision curve analysis across different threshold probabilities. To make easily accessible and applicable of nomogram for clinicians and to patients, mobile based application tool (off line) were developed, that works for both type-1 and Type-2 DM patients to calculate their risk in order to take appropriate intervention measures timely.

Based on study findings, we recommend that for health professionals, who are providing health care services at diabetes mellitus chronic follow up clinic, use the nomogram developed by mobile based application tool to identify high risk of diabetic neuropathy patients for early management and treatment. Additionally Patients use this developed mobile based application tool for their informed choice of treatment by calculating their individualized risk either to keep or improve their health condition. Finally for future researchers improve the model by adding additional predictors like HbA1c, family history of DM and smoking through prospective study design and using monofilament examination by detect enough number of events. Besides, validate externally by using independent sample datasets to be applicable in all health institution.

## Supporting information

**S1 File. Missing variables and its percentage for prediction of diabetic neuropathy among DM patients Northwest Ethiopia, 2005–2021.**
(DOCX)

**S2 File. Predictive performance of for individual and combined predictors by AUC value for prediction of diabetic neuropathy among DM patients North West Ethiopia 2005–2021.**
(DOCX)

## Author Contributions

**Conceptualization:** Negalgn Byadgie Gelaw, Achenef Asmamaw Muche, Adugnaw Zeleke Alem, Nebiyu Bekele Gebi, Yazachew Moges Chekol, Tigabu Kidie Tesfie, Tsion Mulat Tebeje.

**Data curation:** Negalgn Byadgie Gelaw, Achenef Asmamaw Muche, Adugnaw Zeleke Alem, Nebiyu Bekele Gebi, Yazachew Moges Chekol, Tigabu Kidie Tesfie, Tsion Mulat Tebeje.

**Formal analysis:** Negalgn Byadgie Gelaw, Achenef Asmamaw Muche, Adugnaw Zeleke Alem, Nebiyu Bekele Gebi, Yazachew Moges Chekol, Tigabu Kidie Tesfie, Tsion Mulat Tebeje.

**Funding acquisition:** Negalgn Byadgie Gelaw.

**Investigation:** Negalgn Byadgie Gelaw, Achenef Asmamaw Muche, Adugnaw Zeleke Alem, Nebiyu Bekele Gebi, Yazachew Moges Chekol, Tigabu Kidie Tesfie, Tsion Mulat Tebeje.

**Methodology:** Negalgn Byadgie Gelaw, Achenef Asmamaw Muche, Adugnaw Zeleke Alem, Nebiyu Bekele Gebi, Yazachew Moges Chekol, Tigabu Kidie Tesfie, Tsion Mulat Tebeje.

**Resources:** Negalgn Byadgie Gelaw.

**Software:** Negalgn Byadgie Gelaw, Tsion Mulat Tebeje.

**Supervision:** Adugnaw Zeleke Alem, Nebiyu Bekele Gebi.

**Validation:** Negalgn Byadgie Gelaw, Achenef Asmamaw Muche, Adugnaw Zeleke Alem, Nebiyu Bekele Gebi, Yazachew Moges Chekol, Tigabu Kidie Tesfie, Tsion Mulat Tebeje.

**Visualization:** Negalgn Byadgie Gelaw, Achenef Asmamaw Muche, Adugnaw Zeleke Alem, Nebiyu Bekele Gebi, Yazachew Moges Chekol, Tigabu Kidie Tesfie, Tsion Mulat Tebeje.

**Writing – original draft:** Negalgn Byadgie Gelaw.

**Writing – review & editing:** Negalgn Byadgie Gelaw, Achenef Asmamaw Muche, Adugnaw Zeleke Alem, Nebiyu Bekele Gebi, Yazachew Moges Chekol, Tigabu Kidie Tesfie, Tsion Mulat Tebeje.

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
