## [Decision Letter · Decision Letter 0]

16 Jan 2023

PONE-D-22-27723Prediction of the risk for diabetic neuropathy among diabetes mellitus patients at selected referral hospitals, in Amhara regional state Northwest Ethiopia, January, 2005- December 2021; Development and validation of risk prediction modelPLOS ONE

Dear Dr. Gelaw,

Thank you for submitting your manuscript to PLOS ONE. After careful consideration, we feel that it has merit but does not fully meet PLOS ONE’s publication criteria as it currently stands. Therefore, we invite you to submit a revised version of the manuscript that addresses the points raised during the review process.

We look forward to receiving your revised manuscript.

Kind regards,

Jacopo Sabbatinelli, MD, PhD

Academic Editor

PLOS ONE

Journal Requirements:

Reviewers' comments:

Reviewer's Responses to Questions

**Comments to the Author**

1. Is the manuscript technically sound, and do the data support the conclusions?

Reviewer #1: Partly

2. Has the statistical analysis been performed appropriately and rigorously? 

Reviewer #1: Yes

3. Have the authors made all data underlying the findings in their manuscript fully available?

Reviewer #1: Yes

4. Is the manuscript presented in an intelligible fashion and written in standard English?

Reviewer #1: Yes

5. Review Comments to the Author

Reviewer #1: Negalgn Byadgie Gelaw et al. investigated the prediction of the risk for diabetic nephropathy in Ethiopia. In multivariable logistic regression, glycemic control, physical activity, hypertension, alcohol drinking, type of treatment, white blood cells and red blood cells count were statistically significant. Nomogram and mobile based tool were developed and AUC, discrimination performance by bootstrapping method, and classification and regression performance by machine learning were 73.2%, 71.7%, and 70.2%, respectively.

Major comments

1. In lines 241-246, the complete diagnostic criteria should be presented. The thresholds of each diagnostic component should be shown. How many items should be positive for the diagnosis of diabetic neuropathy? What was the diagnostic criteria for the description of diabetic neuropathy in medical record?

2. In lines 161-179, the design of clinical study design is not well described. Since the authors predict the risk for diabetic neuropathy, they should perform the retrospective cohort study. The authors should describe the exposures and observation periods.

3. In page 10 and Table 1, the authors should describe the definition and thresholds of “rural vs urban”, “alcohol drinking”, “physical activity” and “unhealthy diet”.

4. In page 11 and Table 2, did the patients receive the GLP-1 receptor agonists? GLP-1 receptor agonists may be beneficial for the diabetic neuropathy.

5. In lines 408-411, the formula can be applied for the subjects with normal glucose tolerance (NGT), impaired fasting glucose (IFG), and impaired glucose tolerance (IGT)? The authors should use the continuous values rather than descreted values for the clinical parameters.

Minor comments

1. In lines 79, 97, and 138, “burdon” should be “burden”.

6. PLOS authors have the option to publish the peer review history of their article (what does this mean?). If published, this will include your full peer review and any attached files.

Reviewer #1: No

---

## [Author Response · Author response to Decision Letter 0]

17 Feb 2023

1. . In lines 241-246, the complete diagnostic criteria should be presented. The thresholds of each diagnostic component should be shown. How many items should be positive for the diagnosis of diabetic neuropathy? What was the diagnostic criteria for the description of diabetic neuropathy in medical record?

Authors response; we modified as your comment in revised manuscripit in line 241-246 the complete diagnostic criteria for diabetic neuropathy was different corrected in operational definition.

We use clinical criteria and confirmed by different tests, since different type of neuropathies.

The typical clinical sign and symptoms is numbness on hands and feet’s, pain, tingling, paraesthesia , gait ataxia and confirmed by pinprick and temperature examination, touch sensation by 10g monofilament, vibration sense by biothesiometer and ankle reflex

In medical records, the medical doctors written as diabetic neuropathy after they took detailed clinical history, physical examination with advanced imaging technology. 

2. In lines 161-179, the design of clinical study design is not well described. Since the authors predict the risk for diabetic neuropathy, they should perform the retrospective cohort study. The authors should describe the exposures and observation periods.

Authors response; We were correct as in line 161-169 the design of the study is retrospective follow up study since our interest is follow the exposed groups (Diabetes mellitus patients) for 16 years to identify who is at risk of developing diabetic neuropathy and develop clinical model based on the data got from 808 DM patients for early diagnosis and applying early intervention to prevent further complication and death

3. In page 10 and Table 1, the authors should describe the definition and thresholds of “rural vs urban”, “alcohol drinking”, “physical activity” and “unhealthy diet”.

Authors response; The operation definition and threshold of alcohol drinking, physical activity and unhealthy diet were correct and written in operational definition section

4.In page 11 and Table 2, did the patients receive the GLP-1 receptor agonists? GLP-1 receptor agonists may be beneficial for the diabetic neuropathy

Authors response; After diagnosing DM, the patients took GLP-1 agonists medication to prevent diabetic neuropathy and other DM complication such as; retinopathy, nephropathy

5. In lines 408-411, the formula can be applied for the subjects with normal glucose tolerance (NGT), impaired fasting glucose (IFG), and impaired glucose tolerance (IGT)? The authors should use the continuous values rather than descreted values for the clinical parameters.

Authors response; The clinical model formula used for all confirmed/diagnosed Diabetes mellitus patients with or without normal glucose tolerance to identify low or high risk of DM patients to develop diabetic neuropathy. If low risk, then counsel to try to keep their health condition or if high risk to try to improve their health condition by taking drugs accordingly , avoid eating fat contain foods, do physical exercise. Therefore, the formula is very important for all DM patients to predict diabetic neuropathy.

Using clinical parameters as continuous is recommended to prevent loss of information but fore sake of developing mobile based tool and easily applicability, we categorized based on the standard criteria.

 Thank you!!!

---

## [Decision Letter · Decision Letter 1]

20 Feb 2023

PONE-D-22-27723R1

Prediction of the risk for diabetic neuropathy among diabetes mellitus patients at selected referral hospitals, in Amhara regional state Northwest Ethiopia, January, 2005- December 2021; Development and validation of risk prediction model

PLOS ONE

Dear Dr. Gelaw,

Thank you for submitting your manuscript to PLOS ONE. After careful consideration, we feel that it has merit but does not fully meet PLOS ONE’s publication criteria as it currently stands. Therefore, we invite you to submit a revised version of the manuscript that addresses the points raised during the review process.

We look forward to receiving your revised manuscript.

Kind regards,

Jacopo Sabbatinelli, MD, PhD

Academic Editor

PLOS ONE

Journal Requirements:

Reviewers' comments:

Reviewer's Responses to Questions

**Comments to the Author**

1. If the authors have adequately addressed your comments raised in a previous round of review and you feel that this manuscript is now acceptable for publication, you may indicate that here to bypass the “Comments to the Author” section, enter your conflict of interest statement in the “Confidential to Editor” section, and submit your "Accept" recommendation.

Reviewer #1: All comments have been addressed

2. Is the manuscript technically sound, and do the data support the conclusions?

Reviewer #1: Yes

3. Has the statistical analysis been performed appropriately and rigorously? 

Reviewer #1: Yes

4. Have the authors made all data underlying the findings in their manuscript fully available?

Reviewer #1: Yes

5. Is the manuscript presented in an intelligible fashion and written in standard English?

Reviewer #1: No

6. Review Comments to the Author

Reviewer #1: Authors response: After diagnosing DM, the patients took GLP-1 agonists medication to prevent diabetic neuropathy and other DM complication such as; retinopathy, nephropathy.

Please describe the percentage of the patients treated with the GLP-1 receptor agonists in Table 2.

7. PLOS authors have the option to publish the peer review history of their article (what does this mean?). If published, this will include your full peer review and any attached files.

Reviewer #1: No

---

## [Author Response · Author response to Decision Letter 1]

10 Jul 2023

Dear Editors

Manuscript ID: PONE-D-22-27723R1

First of all I would like thank for your expertfull comments to be eligible our work manuscript in your journal and I have one publication by other area and familiar with your PLOS ONE journal policies. 

Response to editors; As your request to amend the title either in the manuscript or online submission form.

The title was amend in the manuscript which is identical in the original manuscript and revised manuscript with track changes as" Development and validation of risk prediction model for diabetic neuropathy among diabetes mellitus patients at selected referral hospitals, in Amhara regional state Northwest Ethiopia, 2005-2021"

The other comments was corrected as response to reviewers and updated to be eligible as PLOS ONE journal guideline 

I was amend in the manuscript at all pages

---

## [Editor Report · Decision Letter 2]

24 Jul 2023

Development and validation of risk prediction model for diabetic neuropathy among diabetes mellitus patients at selected referral hospitals, in Amhara regional state Northwest Ethiopia, 2005-2021.

PONE-D-22-27723R2

Dear Dr. Gelaw,

We’re pleased to inform you that your manuscript has been judged scientifically suitable for publication and will be formally accepted for publication once it meets all outstanding technical requirements.

Kind regards,

Jacopo Sabbatinelli, MD, PhD

Academic Editor

PLOS ONE

Additional Editor Comments:

Extensive language editing is recommended during the proofreading stage.

---

## [Editor Report · Acceptance letter]

1 Aug 2023

PONE-D-22-27723R2 

Development and validation of risk prediction model for diabetic neuropathy among diabetes mellitus patients at selected referral hospitals, in Amhara regional state Northwest Ethiopia, 2005-2021 

Dear Dr. Gelaw:

I'm pleased to inform you that your manuscript has been deemed suitable for publication in PLOS ONE. Congratulations! Your manuscript is now with our production department. 

Kind regards, 

on behalf of

Dr. Jacopo Sabbatinelli 

Academic Editor

PLOS ONE